# Response to Vaccination against Mumps in Medical Students: Two Doses Are Needed

**DOI:** 10.3390/v13071311

**Published:** 2021-07-07

**Authors:** Andrea Trevisan, Angelo Moretto, Chiara Bertoncello, Annamaria Nicolli, Stefano Maso, Maria Luisa Scapellato, Paola Mason

**Affiliations:** Department of Cardiac Thoracic Vascular Sciences and Public Health, University of Padova, via Giustiniani 2, 35128 Padova, Italy; angelo.moretto@unipd.it (A.M.); chiara.bertoncello@unipd.it (C.B.); annamaria.nicolli@unipd.it (A.N.); stefano.maso@unipd.it (S.M.); marialuisa.scapellato@unipd.it (M.L.S.); paola.mason.1@unipd.it (P.M.)

**Keywords:** mumps, vaccination, health care workers, medical students

## Abstract

Mumps is a vaccine-preventable infectious disease diffuse worldwide. The implementation of mumps vaccination reduced largely the spread of infection. On 11,327 Medical School students the prevalence of mumps positive antibodies was evaluated according to dose/doses of vaccine, year of birth and sex. Compliance to mumps vaccine was low in students born before 1990 but increased consistently after this year, above all compliance to two doses, due to the implementation of the vaccine offer. Positivity of mumps antibodies is significantly (*p* < 0.0001) lower in students vaccinated once (71.2%) compared to those vaccinated twice (85.4%). In addition, students born after 1995, largely vaccinated twice, showed a seropositivity near to 90%. Further, females had a significantly (*p* < 0.0001) higher proportion of positive antibodies after vaccination than males, both one (74.6% vs. 64.7%) and two doses (86.8% vs. 82.9%). Finally, seropositivity after two vaccine doses remains high (86.1%) even 15 years after the second dose. In conclusion, the research highlighted that vaccination against mumps reaches a good level of coverage only after two doses of vaccine persisting at high levels over 15 years and induces a more significant response in females.

## 1. Introduction

Mumps is a prevalently benign infectious disease, but several complications can occur [1,2], especially if contracted during adulthood. The basic reproductive rate (R_0_) was calculated to be 10–12, lower than measles, pertussis and varicella in order of infectivity [3]. A vaccine against mumps has been available since 1967.

In Italy, the vaccination against mumps virus was recommended for all males susceptible to mumps since 1982. However, mass vaccination was launched to prevent measles, mumps and rubella (MMR) in 1999 [4]. This allowed a drastic decrease in reported mumps cases from 24,743 in 2000 (43.5 cases/100.000) to 777 in 2018 (1.3 cases/100.000) [5].

The implementation of vaccination against mumps in the world has made it possible to drastically reduce the incidence of the disease even below 0.1 cases/100,000 in certain countries such as the United States [6] and Finland [7,8].

Two Italian serosurveys carried out in 1996–1997 [9] and in 2003–2004 [10] have highlighted similar susceptibility to mumps in age classes 2–14 years and 15–39 years (higher than 20% and 10%, respectively). It is unknown if differences in susceptibility were due to vaccination or past infection.

Further, a seroepidemiology in Western Europe [11] highlighted a high mumps disease incidence in Italy due to a large proportion of negative antibodies related to a poor vaccine coverage; as observed by Eriksen et al. [12], in mumps outbreak countries antibody titers were low.

The current vaccination schedule (MMR) provides two doses of vaccine, between the first and the second years of life, and at six years of age. Since June 2017, MMR has been mandatory in Italy for infants and children 0–16 years of age [13], and is strongly recommended to HCWs by the National Vaccination Prevention Plan 2017–2019 [14] and by the so-called “Pisa Card or Paper” that was drawn up during the work of the National Conference “Medice cura te ipsum” held in Pisa on 27 and 28 March 2017 to promote vaccination practice among health professionals to achieve control of diseases preventable with vaccination [15].

Despite the vaccination coverage with two doses, several cases of mumps recently occurred [16], many more than rubella and measles [2]. This suggests that effectiveness of mumps vaccine is influenced by several factors that are not easy to understand; in fact, after a third dose of mumps vaccine, despite a prompt response, a return to the baseline of antibody level was observed after one year [17].

The aim of the present research was to study the seroepidemiology of mumps in a large population of medical school students, considering that they are the future HCWs. The compliance to mumps vaccination after the introduction of MMR schedule and the presence of mumps antibodies were investigated to understand the best strategy to avoid outbreaks of mumps.

## 2. Materials and Methods

### 2.1. Population

A total of 13,517 students attending to Padua University Medical School (Northern Italy) were submitted to health surveillance during the period 2004–2020, but according to the following enrollment criteria: (1) to be born in Italy and then to be submitted to the same vaccination schedule; (2) availability of a Public Health Office certificate certifying vaccination against mumps, in addition to other vaccines (certificate of this type is believed to be reliable); (3) measurement of antibodies against mumps. A total of 826 were excluded for not having been born in Italy, 1364 because they lacked a valid vaccination certificate and 15 for lacking an antibody titer. Finally, 11,327 students corresponding to the criteria described above were enrolled for the research.

The students attended graduate courses in the health care professions (5926, 52.3%), medicine and surgery (5049, 44.6%) and dentistry (352, 3.1%). There were more females (7388, 65.2%) than males (3939, 34.8%); ratio males/females 0.53. This discrepancy between males and females was more prevalent in health care professions (ratio 0.34) than in medicine and surgery (ratio 0.79) or dentistry (ratio 1.30) courses.

The students were further divided into five year of birth groups (born before 1980, between 1980 and 1985, between 1986 and 1990, between 1991 and 1995, and after 1995) to better highlight the differences in vaccination compliance as previously described [18].

According to the legislative decree 81/08 [19], the workers are subjected to health surveillance (and among the workers they are considered). It is the duty of the occupational physician (according to Italian laws) to draw up the health surveillance protocol which provides the acquisition of all the documents necessary to express an opinion of suitability and to subject the workers to the complementary tests that the occupational physician considers useful. In the case of health care professionals (including students), the protocol also provides the dosage of the antibody titer against the most common communicable infectious diseases.

### 2.2. Antibody Measurement

The measurement of mumps antibodies was qualitative by means of commercial enzyme-linked immunosorbent assay (EIA) Enzygnost (Dade Behring, Marburg, Germany). Equivocal results had been statistically processed as negative according to Center for Disease Control and Prevention (CDC) recommendations [20] In the absence of a well-defined protective serological threshold, EIA remains a critically useful measurement of vaccine immunogenicity [21].

### 2.3. Statistics

Chi-square (χ^2^) test 2 by 2 (Yates’ correction) was used to compare percentage of antibody positivity. Other statistical analyses are descriptive. Significance is stated by *p* < 0.05. Statsdirect 2.7.7 version (Statsdirect Ltd., Birkenhead, Merseyside, UK) has been used for statistical analyses. 

## 3. Results

Of 11,327 students, 23.6% have never been vaccinated, 20.7% were vaccinated once and 55.7% twice. There were no differences according to sex for compliance to vaccination. The compliance increased over the years (Table 1) and students born after 1995 were for the most part vaccinated twice (89.9%), all with MMR.

The positivity to antibodies after two doses was significantly (*p* < 0.0001) higher than after one dose (85.4% and 71.2%, respectively), in males and females. In addition, females showed a significantly higher percentage of positivity (*p* < 0.0001) than males, either after one or two doses. There was no significant difference in positivity between students who were not vaccinated and those receiving just one dose (Table 2). 

It should be noted that positivity in unvaccinated students dramatically decreased according to year of birth in the younger age group (born after 1990). Conversely, exclusing the older age group (born before 1985) where the number of vaccinated (especially with two doses) is low, an increasing rate of positive antibodies was observed in vaccinated students born after 1995. In students vaccinated once and born in 1991–1995, a significant decrease of positive antobodies was observed towards the 1986–1990 born cohort (*p* < 0.0001) and those born after 1995 (*p* = 0.0246). Female students showed prevalently a significantly higher positive response to the vaccine than males (Table 3). Two doses of vaccine were able to yield altogether positive antibodies over 80%, excluding males of the cohort born in 1986–1990 whose percentage of positive antibodies was 76%. 

Waning of antibodies over time was further investigated. Time after either the single or the second vaccine dose was subdivided into four intervals as follows: 0–5, 6–10, 11–15 and over 15 years. A significant drop to 64.8% (*p* < 0.0001) was observed in subjects vaccinated once but not in those vaccinated twice after more than 15 years (Figure 1).

## 4. Discussion

The main results of the present study are: (1) low vaccination compliance to mumps vaccine of students born before 1985; (2) progressive decrease of antibody positivity in unvaccinated students from the year of birth group 1991–1995 (below 50%) and further reductions in students born after 1995 (below 20%); (3) significant differences in response between one (71.2% of positives) or two (85.4% of positives) doses of vaccine; (4) higher positive response to vaccine in females than in males (both one and two doses); (5) antibodies’ persistence at a level near 90% of positivity over fifteen years after two doses of vaccine and only at 64.8% after a single dose.

A weakness of this study is that the antibody measurement is qualitative because the evaluation on antibody titer was not done. In addition, the vaccination certificate does not provide information on the strains used in vaccine formulation. On the other hand, the breadth of the casuistry, the possibility of stratification by year of birth groups and the completeness of the data are certainly strengths.

The large implementation of vaccination against mumps is due to the introduction of the MMR combined vaccine in 1999 [4], since the use of mumps vaccine alone constituted only 2.3% (date of birth 1982–1996) of those vaccinated once and 0.5% (first dose, date of birth 1984–1993) of those vaccinated twice. These students were born (except 2) in Northern Italy. Vaccination coverage at 24 months for mumps was estimated to be 53% in 1998 [9].

The increase of vaccination coverage certainly reduced the spread of wild mumps virus, as demonstrated by the small percentage of unvaccinated subjects with positive antibodies after the implementation of MMR vaccine. The unvaccinated subjects have fewer opportunities to contract the disease and, therefore, the percentage of positivity is very low and the poor circulation of the virus does not allow the vaccinated to have the booster effect that allows the antibody rate to be kept high, since that mumps vaccine-induced immunity has a faster decay due to a lower frequency of memory B cells and lower avidity of IgG [22,23]. Indeed, despite the large vaccination coverage of the population, several outbreaks have been reported in developed countries [7], and in USA cases of mumps recently doubled [16].

In a previously published mumps serosurvey an immune coverage below 80% in medical students was observed [24,25]. One dose of vaccine may not be enough to give an immune coverage. In addition to this, the drop of positive antibodies in students of the 1991–1995 cohort vaccinated once is probably related to the use of the Rubini strain that was introduced in 1991 in the TRIVIRATEN^®^ formulation. The proven ineffectiveness of this vaccine [26,27] led to its subsequent withdrawal and to its replacement with other strains (the Jeryl Lynn strain is the one currently in use). In fact, when the same cohort is vaccinated twice, the positivity exceeds 80 percent; probably, the second dose with another strain (probably Urabe or Jeryl Lynn strain) was sufficient to compensate for the supposed poor efficacy of the Rubini strain. In any case, two doses of the vaccine are necessary to maintain immune coverage above 80 percent, as one dose, even with more effective strains, does not appear to be sufficient. The effectiveness of one dose is low and the antibodies developed after immunization tend to decrease to unprotective levels. Two doses of vaccine can ensure good immune coverage for more than 15 years as opposed to a single dose and could be enough to prevent mumps outbreaks. Nevertheless, within the past years there have been many outbreaks all over the world, even in populations vaccinated twice.

Infection induces a natural immunity that is greater than double that of vaccine-induced immunity [28], but vaccine failure may be related to failed seroconversion (primary vaccine failure), waning immunity (secondary vaccine failure), mismatched vaccination records and vaccine escape due to viral evolution [29,30,31,32,33]. On the other hand, a high vaccine coverage with two doses dramatically reduced cases of mumps in China [34].

A high vaccine coverage is necessary to reach the herd immunity that for mumps is about 90–92% [3]. For this purpose, some authors suggest that a third dose of vaccine is safe and induces at least a short-term benefit for people in outbreak settings [35,36]. 

In Italy, vaccination coverage at 24 months for mumps fell starting from 2013 (90.30%) to reach the lowest percentage in 2015 (85.23%). This was followed by an increase in coverage. The most recent data provided by the ministry of health indicate coverage for two-year-olds of 95.52% and for 18-year-olds of 84.02% with two doses and 89.96% with one [37].

Further, two doses of vaccine allow a stable percentage of positive antibodies over time, near to 90% and then to herd immunity, more than fifteen years after vaccination, supporting the World Health Organization position paper [38]. The importance of mumps vaccination is highlighted by a recent study [39] reporting a hospital outbreak scenario among medical residents with no previous vaccination record against mumps who had a high rate of complications. The study suggests the relevance of analyzes aimed at ascertaining vaccination and immune coverage specifically for mumps.

In a recent serosurvey of Turkish health care workers (HCWs) and students [40], a seropositivity of 74% was found, but vaccination coverage was based on a self-reported questionnaire where about 70% self-reported as vaccinated. These results seemed very similar to those of our casuistry, both among the unvaccinated and those vaccinated with a single dose. The questionnaire does not show with how many doses the HCWs were vaccinated. Finally, sex differences in mumps’ response to vaccination were well recognized in the present study and females are significantly more responsive than males. On the other hand, this evidence was only recently supported [40], but not by another serosurvey [10,41]. Although the innate immunity is similar [42], adaptive immunity is more pronounced in females. The question of whether the female sex responds better to vaccinations is still open, although some evidence supports it for some vaccine types [43,44,45] but not others [46,47]. 

## 5. Conclusions

Compared to measles and rubella, mumps has always received less attention, so much so that wide vaccination coverage was obtained only after the introduction of the combined MMR vaccine. Vaccination coverage with two doses of future HCWs is around 90%, particularly in the younger generations, and seropositivity after two doses of vaccine appears to be high. Thanks to the introduction of the combined MMR vaccine, a careful awareness campaign and, desirably (as for the newborn), the mandatory requirement of a certain number of vaccines for HCWs, will allow accurate control of the transmission of infectious diseases in the hospital setting.

## Figures and Tables

**Figure 1 viruses-13-01311-f001:**
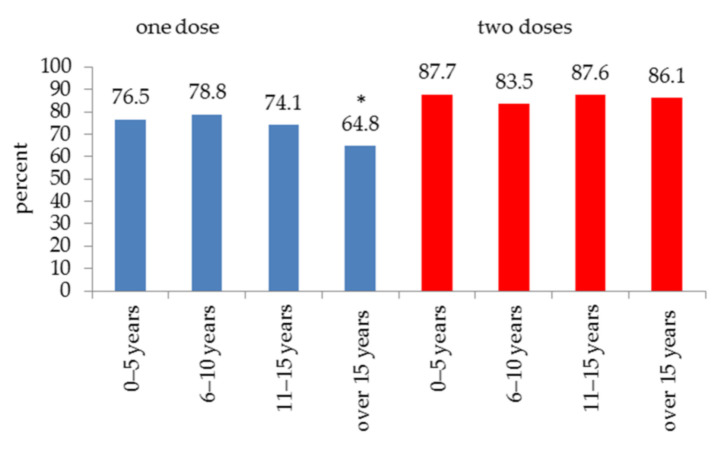
Positivity of mumps antibodies over time after one or two doses of vaccine. A significant (* *p* < 0.0001) waning of antibodies was observed after 15 years but only if one dose of vaccine was administered.

**Table 1 viruses-13-01311-t001:** Vaccination compliance according to year of birth groups.

Year of Birth	*N*.	No Vaccine (*N*.)	%	One Dose (*N*.)	%	Two Doses (*N*.)	%
before 1980	506	492	97.2	12	2.4	2	0.4
1980–1985	1562	1146	73.4	376	24.1	40	2.6
1986–1990	2718	742	27.3	1216	44.7	760	28.0
1991–1995	3927	219	5.6	547	14.6	3161	80.5
after 1995	2614	73	2.8	190	7.3	2351	89.9
all	11,327	2672	23.6	2341	20.7	6314	55.7

**Table 2 viruses-13-01311-t002:** Positivity of mumps antibodies according to vaccine doses and sex. Statistical analysis between doses, males and females are performed by the Chi-square (χ^2^) test with Yates’ correction.

Vaccination	*N*.	Positives	%	χ^2^	*p*	χ^2^	*p*
				differences between doses		differences between sexes	
all students	11,327	8999	79.4				
no vaccine	2672	1937	72.5				
one dose	2341	1667	71.2				
two doses	6314	5395	85.4	229.523	<0.0001		
all males	3939	3000	76.2				
no vaccine	930	651	70.0				
one dose	801	518	64.7				
two doses	2208	1831	82.9	113.346	<0.0001		
all females	7388	5999	81.2			39.625	<0.0001
no vaccine	1742	1286	73.8			4.255	=0.0391
one dose	1540	1149	74.6			24.918	<0.0001
two doses	4106	3564	86.8	119.751	<0.0001	17.018	<0.0001

**Table 3 viruses-13-01311-t003:** Percentage of positive antibodies according to year of birth groups after each dose of vaccine (if received) and sex. Statistical differences between males and females are performed by the chi-square (χ^2^) test with Yates’ correction.

Year of Birth	All Students			Males			Females				
**no vaccine**	***N*.**	**positives**	**%**	***N*.**	**positives**	**%**	***N*.**	**positives**	**%**	**χ^2^**	***p***
before 1980	492	382	77.6	219	160	73.1	273	222	81.3	4.311	=0.0379
1980–1985	1146	909	79.3	351	272	77.5	795	637	80.1	0.874	n.s.
1986–1990	742	536	72.2	236	171	72.5	506	365	72.1	0.000	n.s.
1991–1995	219	96	43.8	87	38	43.7	132	58	43.9	0.000	n.s.
after 1995	73	14	19.2	37	10	27.0	36	4	11.1	2.044	n.s.
**one dose**	***N*.**	**positives**	**%**	***N*.**	**positives**	**%**	***N*.**	**positives**	**%**	**χ^2^**	***p***
before 1980	12	11	91.7	8	7	87.5	4	4	100.0	0.000	n.s.
1980–1985	376	291	77.4	122	86	70.5	254	205	80.7	4.350	=0.0370
1986–1990	1216	897	73.8	378	256	67.7	838	641	76.5	9.898	=0.0017
1991–1995	547	334	61.1	209	108	51.7	338	226	66.9	11.901	=0.0006
after 1995	190	134	70.5	84	61	72.6	106	73	68.9	0.162	n.s.
**two doses**	***N*.**	**positives**	**%**	***N*.**	**positives**	**%**	***N*.**	**positives**	**%**	**χ^2^**	***p***
before 1980	2	2	100.0	1	1	100.0	1	1	100.0	0.000	n.s.
1980–1985	40	39	97.5	22	21	95.5	18	18	100.0	0.000	n.s.
1986–1990	760	637	83.8	262	199	76.0	498	438	88.0	17.344	<0.0001
1991–1995	3161	2630	83.2	1129	931	82.5	2032	1699	83.6	0.687	n.s.
after 1995	2351	2087	88.8	794	679	85.5	1557	1408	90.4	12.249	=0.0005

Legened: χ^2^ and *p* refer to the comparison between males and females.

## Data Availability

Raw data are available upon request to the corresponding author.

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
