# Peer review of "Response to Vaccination against Mumps in Medical Students: Two Doses Are Needed"

_viruses, 2021, doi:10.3390/v13071311_

Round 1

Reviewer 1 Report

Trevisan and colleagues describe a mumps seroprevalence study conducted among a cohort of medical students in Northern Italy with analyses stratified by vaccine dose, age, and biological sex. Routine seroprevalence studies in key subgroups of the population (such as future healthcare workers) are crucial for maintaining mumps surveillance and preventing outbreaks. The authors report that seropositivity was higher among students who received two doses of vaccine (85.4%) compared to those who received a single dose (71.2%). Interestingly, women had a higher overall seropositivity rate than men across the entire cohort, which is consistent with observations in the literature of women exhibiting higher humoral immune responses after vaccination. It is important to note that seropositivity does not equate to protection from mumps disease. There have been numerous reports of mumps outbreaks across the globe among highly immunized populations with many of the infected having received two doses of MMR vaccine in childhood. While this report has value for the literature, the authors must address the comments below before I recommend this work for publication.   

Minor Comments:

  1. Were samples yielding equivocal results repeated, or were they assigned as equivocal based on a single test?
  2. Since the cohort was predominantly female, was there sufficient statistical power for sex-based comparisons? Provide a statement supporting the level of statistical power.

Major Comments:

  1. It is interesting (although not all that unexpected) that the unvaccinated individuals had the same level of seropositivity as those receiving 1 dose of vaccine. How many of the unvaccinated individuals have a medical history of mumps as children?
  2. While seropositivity is informative, it only indicates that antibody titers are above a certain level. It would be more informative to attempt to quantify this in some manner. It would be interesting to see ELISA OD values (which should approximate the antibody titer) plotted for the three groups (unvaccinated, 1 dose, 2 dose) to have some idea of what the antibody levels were between groups.
  3. It would also be interesting to see how the actual antibody titer changes across time by plotting the OD values to get a better idea of waning immunity. I suspect that you’ll find antibody levels/OD values are decreasing over time even though individuals may remain seropositive.    

Author Response

Comments and Suggestions for Authors

Trevisan and colleagues describe a mumps seroprevalence study conducted among a cohort of medical students in Northern Italy with analyses stratified by vaccine dose, age, and biological sex. Routine seroprevalence studies in key subgroups of the population (such as future healthcare workers) are crucial for maintaining mumps surveillance and preventing outbreaks. The authors report that seropositivity was higher among students who received two doses of vaccine (85.4%) compared to those who received a single dose (71.2%). Interestingly, women had a higher overall seropositivity rate than men across the entire cohort, which is consistent with observations in the literature of women exhibiting higher humoral immune responses after vaccination. It is important to note that seropositivity does not equate to protection from mumps disease. There have been numerous reports of mumps outbreaks across the globe among highly immunized populations with many of the infected having received two doses of MMR vaccine in childhood. While this report has value for the literature, the authors must address the comments below before I recommend this work for publication.   

We are grateful to the reviewer for his suggestions. Point by point:

Minor Comments:

  1. Were samples yielding equivocal results repeated, or were they assigned as equivocal based on a single test?

Reply: As explained below, the analyses are performed by the clinical microbiology laboratory.

  1. Since the cohort was predominantly female, was there sufficient statistical power for sex-based comparisons? Provide a statement supporting the level of statistical power.

Reply: The study analyzes data from the entire student population attending graduate courses in the health care professions: 11,327 students.

Major Comments:

  1. It is interesting (although not all that unexpected) that the unvaccinated individuals had the same level of seropositivity as those receiving 1 dose of vaccine. How many of the unvaccinated individuals have a medical history of mumps as children?

Reply: In general, everyone remembered common childhood exanthematic diseases (especially chickenpox and measles), while they were unsure about mumps. This disease can occur, especially in children, with mild symptoms or symptoms similar to morbidity of the upper respiratory tract or middle ear.

  1. While seropositivity is informative, it only indicates that antibody titers are above a certain level. It would be more informative to attempt to quantify this in some manner. It would be interesting to see ELISA OD values (which should approximate the antibody titer) plotted for the three groups (unvaccinated, 1 dose, 2 dose) to have some idea of what the antibody levels were between groups.

Reply: We agree with the reviewer that it would be interesting to know the OD value of each positive sample. Unfortunately, knowing this data is impossible, both for the size of the sample (which would be the least), but mainly because the analyses are performed by the clinical microbiology laboratory and not in our laboratory.

  1. It would also be interesting to see how the actual antibody titer changes across time by plotting the OD values to get a better idea of waning immunity. I suspect that you’ll find antibody levels/OD values are decreasing over time even though individuals may remain seropositive.

Reply: The answer to this request, which is of great interest, is similar to the previous one.   

Reviewer 2 Report

INTRODUCTION

Lines 50-2, suggest leave this statement about Turkish HCW to the Discussion.

Line 61, explain what the Pisa Card is

MATERIALS AND METHODS

This section needs further information to enable readers to determine whether any biases in subject selection exist in the study.

Line 77, explain (a) who issues the Public Health Office certificate; (b) whether it is mandatory for HC students to obtain a certificate indicating their mumps vaccination status on entry to Medical School, (c) if so, under what policy or legislation.

Line 79-80, provide detailed information on the law that requires HC students to give blood for mumps serology, and how the law is put into effect at the Medical School; has the same law been in place since 2004?

2.2 Antibody measurement, add a sentence saying that the mumps Enzygnost IgG assay is thought to correlate with protective immunity and give a reference

RESULTS

Provide information on the number of HC students actually enrolled at the Medical School in 3 time periods, eg 2004-2010, 2011-2015 and 2016-2020, who were Italian born, and then for each time period, the number of students who had a Certificate and the number in whom serology was available. This will enable the reader to judge how complete the data are and whether there is any chance of selection bias.

Author Response

Comments and Suggestions for Authors

We are grateful to the reviewer for his suggestions. Point by point:

INTRODUCTION

Lines 50-2, suggest leave this statement about Turkish HCW to the Discussion.

Reply: as requested, the statement has been moved to the Discussion.

Line 61, explain what the Pisa Card is

Reply: Pisa card was explained in the text. In any case, the Pisa card was drawn up during the work of the National Conference "Medice cura te ipsum" held in Pisa on 27 and 28 March 2017 to promote vaccination practice among health professionals to achieve control of diseases preventable with vaccination.

MATERIALS AND METHODS

This section needs further information to enable readers to determine whether any biases in subject selection exist in the study.

Reply: All cases have been entered and which have been excluded.

Line 77, explain (a) who issues the Public Health Office certificate; (b) whether it is mandatory for HC students to obtain a certificate indicating their mumps vaccination status on entry to Medical School, (c) if so, under what policy or legislation.

Line 79-80, provide detailed information on the law that requires HC students to give blood for mumps serology, and how the law is put into effect at the Medical School; has the same law been in place since 2004?

Reply: To these two suggestions we provide a single answer. According to the legislative decree September 19, 1994, number 626 and then to the legislative decree April 9, 2008, number 81, the workers are subjected to health surveillance (and among the workers they are considered interns in 626/94 and students in 81/08). It is the duty of the occupational physician (according to Italian laws) to draw up the health surveillance protocol which provides the acquisition of all the documents necessary to express an opinion of suitability and to subject the workers to the complementary tests that the occupational physician considers useful. In the case of healthcare professionals (including students), the protocol provides the dosage of the antibody titre against the most common communicable infectious diseases also.

2.2 Antibody measurement, add a sentence saying that the mumps Enzygnost IgG assay is thought to correlate with protective immunity and give a reference

Reply: the requested sentence and reference have been added.

RESULTS

Provide information on the number of HC students actually enrolled at the Medical School in 3 time periods, eg 2004-2010, 2011-2015 and 2016-2020, who were Italian born, and then for each time period, the number of students who had a Certificate and the number in whom serology was available. This will enable the reader to judge how complete the data are and whether there is any chance of selection bias.

Reply: requested information have been added (partially) in the text (in material and methods). Of those enrolled in medicine, a number were subsequently revised as a resident, a very small number enlisted in the ranks of researchers. As for the health professions, some have attended masters, but most have dispersed. For your interest:

total N.

born in Italy N.

no born in Italy N.

availability of the certificate N.

born in Italy N.

no born in Italy N.

no antibody measure

2004-2010

5697

5276

421

4337

4306

31

13

2011-2015

4748

4526

222

4332

4254

78

2

2016-2020

3072

2889

183

2875

2767

108

0

13,517

12,691

826

11,544

11,327

218

15

Round 2

Reviewer 2 Report

Thank you for your changes to the manuscript and your additional responses. From the viewpoint of readers interested in the public health policies which in part explain your findings, I would recommend that you include in Methods a reference to legislative decree 81/08 which requires vaccine preventable disease surveillance of health care students.

RESULTS

in the 2nd last line of Table 3, in which you present the antibody prevalence by sex for students born in 1991-95 you state that a difference between 82.% in males and 83.6% in females is highly statistically different. However, when i conducted a quick calculation of the X2 and p value i found that there was no significant difference. This may be the only error of this type on your part, but i would ask you to recheck all the statistical calculations prior to resubmission.

DISCUSSION

The discussion is missing context which would be provided by data showing the change over time in mumps vaccination coverage in Italian children. Is this information available? You use reference 4 to state that 2.9% ahd received one dose of mumps vaccine and 0.5% had received a 2nd dose - please add the year this coverage relates to and what population (ie what geographic location and what age group(s) it applied to. If you could add any other reference showing mumps vaccine coverage among children (or adults) in Italy (or part of Italy) since that study it would add to the Discussion.

The CONCLUSIONS are appropriate.

Author Response

We are grateful to the reviewer for his suggestions which are sure to make improvements to the manuscript.

Thank you for your changes to the manuscript and your additional responses. From the viewpoint of readers interested in the public health policies which in part explain your findings, I would recommend that you include in Methods a reference to legislative decree 81/08 which requires vaccine preventable disease surveillance of health care students.

Reply: As requested, the legislative reference has been inserted (lines 100-107).

RESULTS

in the 2nd last line of Table 3, in which you present the antibody prevalence by sex for students born in 1991-95 you state that a difference between 82.% in males and 83.6% in females is highly statistically different. However, when I conducted a quick calculation of the X2 and p value is found that there was no significant difference. This may be the only error of this type on your part, but I would ask you to recheck all the statistical calculations prior to resubmission.

Reply: We are grateful to the reviewer for the clarification. There was indeed a glaring error which has been corrected. We have double checked all the data and found a typo (irrelevant for the significance) in Table 2, promptly corrected. I apologize for the big mistake.

DISCUSSION

The discussion is missing context which would be provided by data showing the change over time in mumps vaccination coverage in Italian children. Is this information available? You use reference 4 to state that 2.9% had received one dose of mumps vaccine and 0.5% had received a 2nd dose - please add the year this coverage relates to and what population (i.e. what geographic location and what age group(s) it applied to. If you could add any other reference showing mumps vaccine coverage among children (or adults) in Italy (or part of Italy) since that study it would add to the Discussion.

Reply: Reference 4 is the Ministry of Health circular that launched mass vaccination against measles, mumps and rubella with the introduction of the trivalent vaccine. The numbers reported refer to the small minority of students who had either a single dose of the mumps vaccine alone (2.9%) or the first dose of mumps alone and the second dose of the trivalent vaccine (0.5%) (lines 184-185). The first students were born between 1982 and 1990 (although two were born in 1995 and 1996) and the second between 1984 and 1993. Apart from two, they are all students born in Northern Italy. These data were included in the discussion.

The requirements of the first and last recommendations have been added and referenced.

The CONCLUSIONS are appropriate.

This manuscript is a resubmission of an earlier submission. The following is a list of the peer review reports and author responses from that submission.

Round 1

Reviewer 1 Report

In the current manuscript, Trevisan et al. describe the vaccination compliance to vaccination recommendations of Italian students. Furthermore, the authors describe the percentage of these students exhibiting antibodies to the mumps virus. The data are stratified to the number mumps vaccines, sex, and the year of birth (five-year periods).

Major points

The weakness of the manuscript is that it contains no new information. Moreover, it is not clear whether the data presented herein have been published previously (Vaccines 2020, 8(1), 104; https://doi.org/10.3390/vaccines8010104). The authors should describe what is different to the previous study. It is also essential to describe the period in which the study was performed.

The manuscript contains various auto-citations while important results of other groups are neglected, also those from other Italian groups, e.g. Gabutti et al., 2008, Vaccine 26, 2906–2911. However, many more important studies were ignored.  

The authors frequently use “medical slang” not appropriate for a scientific manuscript. All in all the manuscript is preliminary but might be interesting for a national journal after redrafting.

Minor points:

In scientific reports, results should be described in past tense. However, in the whole manuscript the authors skip between presence and past tense.

Abstract

Page 1, line 21: “In addition, student born after… “

Please add “s” to students

Page 1, line 24: “…higher positive response to vaccine…”

What is a positive response? Few side effects? Please try to state explicitly.

Introduction

Page 2, line 35: “…was calculated to be 10‐12, then lower that measles,…”

Please rephrase or use “than” instead of “then”.

Page 2, 38: “… In Italy, the vaccination against mumps was implemented later, since only in 1982

38 the Health Ministry recommended its administration.”

This sentence indicates that vaccination against mumps was not recommended after 1982. In Italy, vaccination against mumps virus was recommended for all males susceptible to mumps since 1982.

Please rephrase sentence.

Page 3, line 85: “…antibody positivityOther statistical…”

Please add full stop after “positivity”.

Results

Page 4, lines 107 -109: “Conversely, a part the older age group (born before 1985) where the number of vaccinated is low, a tendency to increase of positivity was observed in vaccinated students born after 1995.”

This sentence is confusing. Please rephrase. Try to avoid the term “increase of positivity”.

Page 4, line 110: “… a drop of positivity…”

Please replace this term.

Page 4, line 113: “Two doses of vaccine are able to maintain positivity over 80 percent.”

In Table 3 (two doses) percentage of males born between 1986 and 1990 exhibiting mumps virus antibodies was 76%. Please specify statement.

Page 2 and 4, Table 2 and 3: “Positivity of mumps antibodies …”

Please rephrase

Discussion

Page 5, lines 131 – 133: “2) progressive decrease of unvaccinated students with positive antibodies starting by the year of birth group 1991‐1995 (below 50%) and further reduced in students born after 1995 (below 20%)”

Please rephrase

Page 5, lines 135-136: “5) positive antibodies persistence at level near 90% over fifteen years after two doses of vaccine and only 64.8% after a single dose.”

Please rephrase or delete “positive”

Page 5, lines 140-142: “… the possibility of stratification by year of birth and the completeness of the data are certainly strengths”

Although the authors had the possibility to stratify by year of birth data were stratified by 5-years intervals.

Page 5, lines 143-145: “The large implementation of vaccination against mumps is due to the introduction of MMR combined vaccine in 1999, since the use of mumps vaccine alone was in only

145 2.3% of vaccinated once and 0.5% (first dose) of vaccinated twice.”

Please add reference.

Page 5, line 15-153: “.. since it is known that the antibody response to the mumps virus decreases over time even in naturally immunized subjects [13].”

I did not find this statement in the manuscript of Fiebelkorn et al. (2014).

Page 5, line 158: “…the drop of positivity in students…”

Please rephrase.

Page 5, line 62: “Infact…”

Please insert blank: In fact…

Page 6, line 170: “…could be enough to prevent mumps outbreaks.”

Nevertheless, within the past years there were many outbreaks all over the world even in populations vaccinated twice.

Page 6, lines 177-178: “For this purpose, some authors recommended a third dose of vaccine to

178 prevent mumps outbreaks [25].”

The sentence suggests that application of a third dose per se is helpful to prevent outbreaks. In fact, a third dose is recommended in outbreak situations. Please specify statement.